# Sustainable Surface Water Storage Development: Measuring Economic Benefits and Ecological and Social Impacts of Reservoir System Configurations

**Nishadi Eriyagama [1,*], Vladimir Smakhtin [2] and Lakshika Udamulla [3]**

1   International Water Management Institute (IWMI), Colombo P.O. Box 2075, Sri Lanka
2   UNU Institute for Water Environment and Health, 204–175, Longwood Road South,
    Hamilton, ON L8P-0A1, Canada; vladimir.smakhtin@unu.edu
3   Faculty of Engineering Technology, Open University of Sri Lanka, Nawala P.O. Box 21, Nugegoda, Sri Lanka;
    laudu@ou.ac.lk
*     Correspondence: n.eriyagama@cgiar.org

**Abstract:** This paper illustrates an approach to measuring economic benefits and ecological and social impacts of various configurations of reservoir systems for basin-wide planning. It suggests indicators and examines their behavior under several reservoir arrangement scenarios using two river basins in Sri Lanka as examples. A river regulation index is modified to take into account the volume of flow captured by reservoirs and their placement and type. Indices of connectivity illustrate that the lowest river connectivity in a basin results from a single new reservoir placed on the main stem of a previously unregulated river between the two locations that command 50% and 75% of the basin area. The ratio of the total affected population to the total number of beneficiaries is shown to increase as the cumulative reservoir capacity in a river basin increases. An integrated index comparing the performance of different reservoir system configurations shows that while results differ from basin to basin, the cumulative effects of a large number of small reservoirs may be comparable to those with a few large reservoirs, especially at higher storage capacities.

**Keywords:** surface water storage; water reservoirs; sustainability; flow regulation; river connectivity; equity

## 1. Introduction

Global artificial surface water storage facilities include 60,000 large dams [1] and 800,000 smaller dams [2]. They act as buffers against seasonal and annual variations in water supply for domestic, agricultural and industrial users. At least 43% of global irrigation supplies are provided by reservoirs with a storage capacity larger than 0.1 km³ [3]. Hydropower accounts for 16–17% of global electricity generation [4,5], a significant proportion of which is produced by large dams with a height of 15 m or greater or a capacity of 3 million m³ or greater [1]. Water storage also reduces flood risk and offers recreation, fishing and other services. Given the projected population and economic growth, changes in diets and impacts of climate change, further growth in global surface water storage is likely for the foreseeable future [6,7] although at a slower pace than 50 to 70 years ago [8].

While delivering significant benefits, water storage has environmental and social costs. Flow regulation and river fragmentation are known to create negative ecological consequences on aquatic ecosystems and fisheries dependent on free-flowing rivers [9–12]. Large dams displace communities [13,14], affect the well-being of those living downstream [15,16] and frequently create conflicts between different groups of stakeholders [17,18]. These impacts are contingent on storage capacity, the number of reservoirs and the configuration of a reservoir system in a basin. To better account for both positive and negative cumulative effects, water storage infrastructure development must move away from the practice of ad hoc, single-site planning to basin-wide planning [13,19].

A previous paper by Eriyagama et al. [20] illustrated how an overarching sustainable storage development pathway and acceptable limits to surface storage development can be identified for a given river basin to limit cumulative withdrawals from reservoir storage to a sustainable level. This paper illustrates another aspect of basin-wide storage planning: measuring economic benefits and ecological and social effects of reservoir system configurations consisting of different numbers, storage capacities, modes of siting and spatial distribution of reservoirs. The paper uses the same example case study basins and the same reservoir system configurations as Eriyagama et al. [20].

The body of literature on optimizing operations of existing reservoirs to meet economic and ecological objectives is quite significant [21–26]. However, these studies are limited to an assessment of operation scenarios of existing reservoirs and do not consider basin-wide water storage planning. Trade-offs between economic and ecological objectives in sizing and siting reservoirs in a river basin have been examined in a literature review by Jager et al. [27] and an optimization modeling exercise by Roozbahani et al. [28], but those studies do not comprehensively address social objectives, which are the third most important aspect in storage planning.

This paper illustrates an approach to assessing trade-offs between water storage benefits and ecological and social objectives under different arrangements of water supply reservoirs by investigating the behavior of five indicators measuring economic benefits and ecological and social effects. These indicators were selected after an extensive literature search because they are (a) highly responsive to variations in storage capacity, number, location, and spatial distribution of reservoirs as elaborated by Eriyagama et al. [29] and (b) allow easy acquisition of data for the example case study basins described in Section 3.1 below. The reservoir system configurations were subsequently compared based on the behavior of these indicators.

## 2. Methodology

### 2.1. Measuring Economic Benefits

Annual reservoir yield is the average quantity of water that can be supplied in a year for irrigation, domestic and industrial needs and is generally used to measure the economic benefits from a reservoir. The safe yield is defined as the annual yield that can be supplied at 100% reliability for a given flow record from a reservoir of a given capacity which starts full and refills at least once after the worst drought on record [30]. The safe yield (water supply yield in this paper) is used to measure the economic benefits of water storage under different reservoir system configurations since it is sensitive to changes in the storage capacity, number and spatial distribution. Although supply reliability figures in the range 95–99% are generally used in water resources planning, water supply yield (WS Yield) with 100% reliability was used in this research as an indicator to enable comparisons between different reservoir system configurations. The reliability of the supply does not affect the results of the comparison since the same reliability is used to assess all configurations.

The maximum possible WS Yield from each reservoir was estimated by applying the sequent peak algorithm [31,32] on the time series of inflows to each reservoir while not allowing any specific environmental flow releases other than spills (see Eriyagama et al. [20] for details of the estimation procedure). WS Yields ($m^3$ $year^{-1}$) of individual reservoirs were aggregated to estimate the cumulative yield from the entire reservoir system.

### 2.2. Measuring Ecological Impacts

Indicators for environmental flow yield, flow regulation and river connectivity were used to measure the ecological effects of surface storage development. Sections 2.2.1–2.2.3 explain why these indicators were selected and the calculation formulas, including adjustments made to the original formulas to suit the current research.

### 2.2.1. Environmental Flow Yield

Environmental flow (EF) refers to the quantity and timing of water flows required to sustain freshwater ecosystems and the human livelihoods that depend on those ecosystems. This research used EF Yield to refer to the annual quantity of EF that can be maintained below a given reservoir and is essentially the difference between the mean annual runoff (MAR) at the reservoir location and the WS Yield. Ideally, EF releases are required to be expressed as a variable monthly flow regime in line with natural flow variability. However, since the objective of this research was to compare reservoir system configurations, the indicator annual EF Yield, which is compatible with the indicator annual WS Yield, was used to illustrate differences between configurations in annual flow releases downstream of reservoirs. The cumulative yield for the entire basin was estimated by aggregating individual reservoir EF Yields. Like WS Yield, EF Yield ($m^3$ $year^{-1}$) is also sensitive to changes in storage capacity, number and spatial distribution of reservoirs.

### 2.2.2. Flow Regulation

The proportion of a river's MAR captured by storage reservoirs approximately represents the degree of impact of storage infrastructure on downstream flows [33]. This proportion is commonly referred to as flow regulation [34,35] or the degree of flow regulation (DOR) [33]. Grill et al. [19] further refined the DOR to formulate a river regulation index (RRI) to quantify the overall impact of multiple reservoirs in a basin using a single index value. This is calculated by first weighting the DOR of each individual river segment ($DOR_i$) with its corresponding river volume and then aggregating the results for the entire basin (Equations (1) and (2)).

$$DOR_i = \frac{\sum_{j=1}^{n} S_j}{MAR_i} \tag{1}$$

$$RRI = \sum_{i=1}^{N} DOR_i \frac{rv_i}{V} \tag{2}$$

where $n$ = total number of reservoirs on river segment $i$, $S_j$ = capacity of reservoir $j$ on river segment $i$, $MAR_i$ = mean annual runoff of river segment $i$, $rv_i$ = volume of river segment $i$, $V$ = volume of the entire river network and $N$ = the total number of river segments in the network (a river segment is equivalent to that part of a river contained within a given sub-basin). The DOR and RRI are measured in years.

This weighting system ensures that higher weights are assigned to individual DORs of higher-order river segments over those of lower-order segments, recognizing that river volumes typically increase downstream. As a result, the relative impact on the overall index of dams located on larger (higher-order) rivers or main stems is higher. However, the DOR (Equation (1)) or the RRI (Equation (2)) does not adequately distinguish between whether storage capacity exists as a single large reservoir or as a cluster of small reservoirs since only the aggregated capacity in a given river segment is considered. Therefore, an alternative index was formulated to obtain the overall degree of regulation of a network of reservoirs (referred to hereafter as the modified RRI or MRRI). The intention was to:

(a) Overcome the nonavailability of river volumes (or even river depths and widths) for all river segments of the case study basins and introduce an alternative parameter to weight the individual DOR by each reservoir;

(b) Develop a metric that captures differences between having centralized versus distributed reservoirs in a river network;

(c) Formulate an index capable of demonstrating the differences between reservoirs located in upstream reaches and main stems of rivers.

In the case of a river basin with $n$ number of individual reservoirs,

$$MRRI = \sum_{i=1}^{n} \frac{s_i}{Q} \times \frac{q_i}{Q} \tag{3}$$

where $s_i$ = capacity of reservoir $i$, $q_i$ = cumulative natural MAR at the location of a reservoir $i$, $Q$ = the MAR of the entire river and $n$ = the number of reservoirs.

In Equation (3), the individual DOR for each reservoir is weighted by that fraction of the total MAR that flows past it and is denoted by ($\frac{q_i}{Q}$). This research calls the ratio $\frac{q_i}{Q}$ a distance factor, which ensures that reservoirs located on main stems receive higher weights than those located on upstream reaches. Furthermore, Equation (3) modifies the cumulative DOR by n number of reservoirs ($\sum_{i=1}^{n} \frac{S_i}{Q}$), by a factor equal to $\frac{1}{Q} \times \frac{\sum_{i=1}^{n} S_i q_i}{\sum_{i=1}^{n} S_i}$. This factor is called the distribution factor, which essentially measures the degree to which reservoir capacity is distributed within a river basin as opposed to being lumped together. For instance, a lower distribution factor indicates that reservoir capacity exists as a large number of smaller reservoirs while a larger value indicates otherwise. While the MRRI can have values greater than 1, the distance factor and the distribution factor vary between 0 and 1. The MRRI is sensitive to the storage capacity, location and spatial distribution of reservoirs and is measured in years.

### 2.2.3. River Connectivity

River networks receive nutrient inputs through lateral connectivity between river reaches and their catchments. River networks also provide longitudinal connectivity for the flow of water, nutrients, energy and organisms within a river basin [36–39]. Dams, reservoirs and diversions act as barriers causing river networks to fragment longitudinally and laterally [19,33,35], resulting in the loss of connectivity between different segments of a river and between the river and the sea (Figure 1). Fragmentation degrades the health of river ecosystems, often leading to the decline in migratory fish species [12,35,40,41].

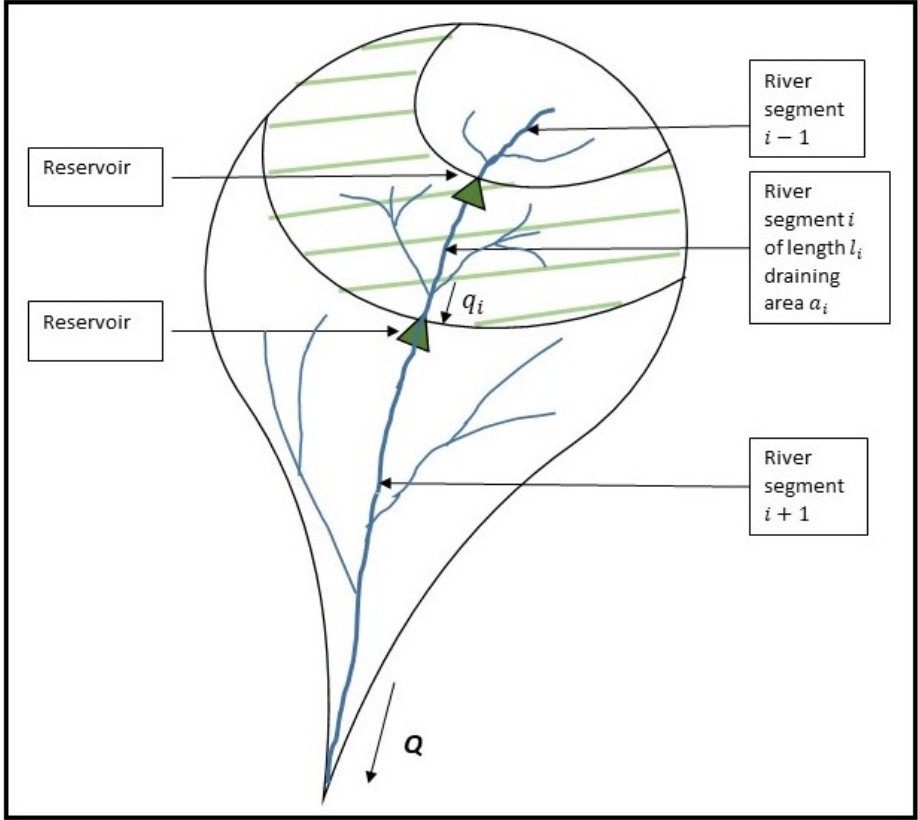

**Figure 1.** Illustration of disconnected river segments created by reservoirs. Q is the mean annual runoff (MAR) of the entire river at the sea outlet while $q_i$ is the MAR at the outlet of river segment $i$.

Several methods are available to measure the degree of river fragmentation and its reciprocal river connectivity, ranging from the number of dams within a basin [42] to

species distribution models (e.g., Lassalle et al., [43]). A method of intermediate complexity is the dendritic connectivity index (DCI) developed by Cote et al. [44]. This research used a derivative of the DCI (as elaborated below) as a measure of river connectivity under different reservoir system configurations. Although less complex than species models, it is still able to capture aspects of structural connectivity in river networks.

The DCI has two subindices, $DCI_{internal}$ measuring the degree of connectivity between different segments of a river (internal connectivity) and $DCI_{external}$ that measures the degree of connectivity between the river and the sea (external connectivity). The DCI is assessed in terms of the probability that fish can move between two randomly selected points in a river network. Therefore, it depends on the number of barriers between the two points and the permeability of these barriers [44]. It is assumed that pairs of river segments share a connectivity value and the weighted average of the connectivity value of all river segment pairs is expressed as the $DCI_{internal}$. The weights are based on the ratio of the lengths of the pair of river segments to the length of the entire river network. Similarly, $DCI_{external}$ is calculated as the probability that a fish can move both upstream and downstream between the mouth of the river and another part of the river network (see Cote et al. [44] for full details of the derivation of the two indices). As shown by Cote et al. [44], when entirely impermeable barriers such as dams are present between river segments, the two indices simplify to Equations (4) and (5).

$$DCI_{internal} = \sum_{i=1}^{n} \frac{l_i^2}{L^2} \times 100 \qquad (4)$$

$$DCI_{external} = \frac{l_s}{L} \times 100 \qquad (5)$$

where $n$ = the total number of disconnected river segments due to the placement of barriers, $l_i$ = the total length of all tributaries of disconnected river segment $i$, $l_s$ = the total length of all tributaries of the river segment directly connected with the sea, $L$ = the total length of the entire river network, $DCI_{internal}$ = internal connectivity and $DCI_{external}$ = external connectivity. An entirely connected (natural) network has an index value of 100, which decreases as barriers are added to the network.

Grill et al. [19] derived an improved internal connectivity index ($RCI_{internal}$) by replacing the river length in $DCI_{internal}$ with the river volume to introduce higher weights to barriers placed in the main stems of rivers over those placed in upstream reaches. Based on the same principle and following reasons a and c in Section 2.2.2, two modified indices ($MRCI_{internal}$ and $MRCI_{external}$) were formulated to measure the internal and external connectivity of a river network (Equations (6) and (7)).

$$MRCI_{internal} = \sum_{i=1}^{n} \frac{a_i^2}{A^2} \cdot \frac{q_i}{Q} \times 100 \qquad (6)$$

$$MRCI_{external} = \frac{a_s}{A} \times 100 \qquad (7)$$

where $n$ = the total number of disconnected river segments due to the placement of barriers, $a_i$ = the basin area drained by river segment $i$, $a_s$ = the basin area of the river segment directly connected to the sea, $A$ = the area of the entire river basin, $q_i$ = MAR at the outlet of river segment $i$ and $Q$ = MAR at the basin outlet to the sea. A fully connected river network has an index value of 100 which decreases as reservoirs are added.

In Equations (6) and (7), basin area is used as a proxy for length as it is easier to estimate. In $MRCI_{internal}$, the indicator based on basin area is subsequently weighted by that fraction of the total MAR that flows out at the end of the river segment ($\frac{q_i}{Q}$) before aggregating. This weighting ensures that barriers on the main stems of rivers receive higher weights than those on upstream reaches. $MRCI_{internal}$ is sensitive to the location and

spatial distribution of reservoirs while $MRCI_{external}$ is sensitive to the location of the most downstream reservoir in the river network. The $MRCI_{internal}$ and $MRCI_{external}$ are unitless.

### 2.3. Measuring Social Impacts

Example indicators designed to measure social equity aspects of water storage development include social efficiency of hydropower [45], social employment efficiency [46] and normalized externality [13]. The normalized externality (NE) is directly relevant to water supply storage and responds well to differences in reservoir configurations; hence, it was used here to assess equity aspects.

NE is the ratio between the number of all affected actors of a water storage project and the number of positively affected actors. Affected actors are benefactors, direct recipients and indirect recipients (Figure 2). Benefactors include the sum of those living in the upstream basin area that drains into the reservoir and those displaced by the reservoir. This is based on the reasoning that owners (or land users) in the upstream basin area and those displaced by the reservoir may not directly benefit from the reservoir but may be constrained by the need to not capture the runoff generated in their area. Instead, they need to allow the runoff to flow downstream to the reservoir; i.e., benefactors provide a service to the reservoir [13]. Direct recipients are the group of people that directly benefit from the reservoir (usually living in the reservoir's command area). Indirect recipients are those located further downstream and subject to the effects of an altered flow regime. It is acknowledged that direct recipients can be present even outside the command area (e.g., upstream of the reservoir and outside the basin altogether) and indirect recipients may still be beneficiaries in the context of hydropower reservoirs. However, this study focuses on consumptive uses only and assumes that all supplies are provided by gravity. Hence the original context and formulation of the indicator by [13] are used (Figure 2).

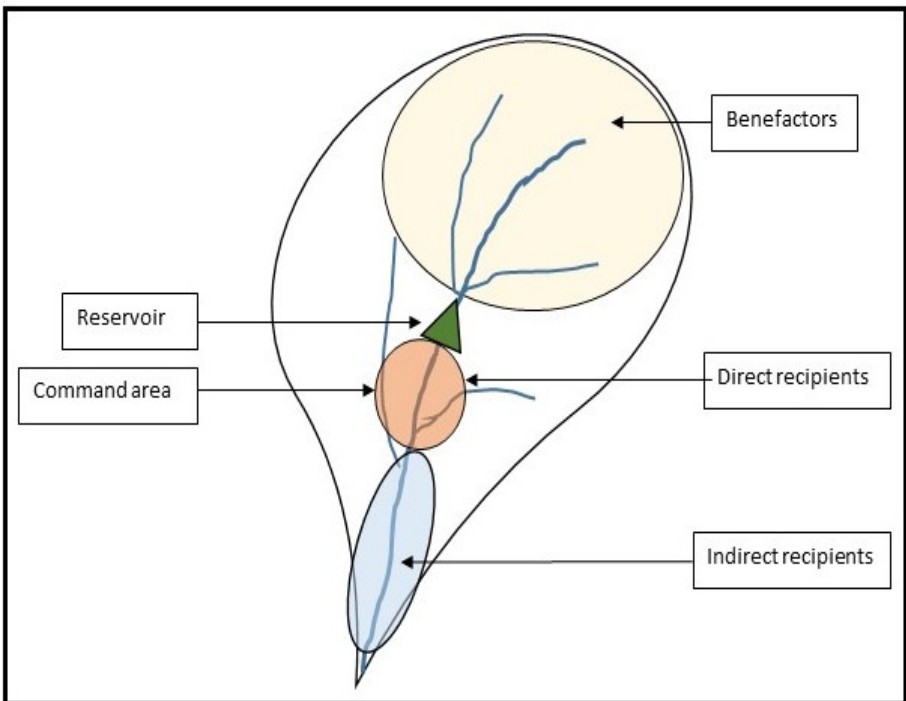

**Figure 2.** Benefactors, direct recipients and indirect recipients of a reservoir project (based on [13]).

The value of NE is always greater than 1. However, the closer its value is to 1, the higher the project is on the equity dimension.

$$\text{NE} = \frac{n_b + n_{dr} + n_{ir}}{n_{dr}} \tag{8}$$

where $n_b$ = number of benefactors, $n_{dr}$ = number of direct recipients and $n_{ir}$ = number of indirect recipients of the storage project.

If there is more than one reservoir in a basin, an overall NE can be estimated considering all actors. Since it was not possible to gather explicit data on $n_b$, $n_{dr}$ and $n_{ir}$ for the reservoir configurations assumed in this research (Section 3), methods were formulated to estimate them as a fraction of the total population in a basin. These formulations were based on the assumption that every individual in a basin has an equal chance of being either a benefactor, a direct recipient or an indirect recipient. This is an idealized situation since direct recipients generally comprise a target group served by the purpose of the reservoir rather than a random fraction of the total basin population. However, given that the research focuses on water supply for multiple needs ranging from irrigation to domestic and industrial use, it is reasonable to assume that every individual has an equal chance of benefiting and being affected by the reservoirs. The estimation procedure is shown by Equations (9) to (12) and is further illustrated in Figure 3.

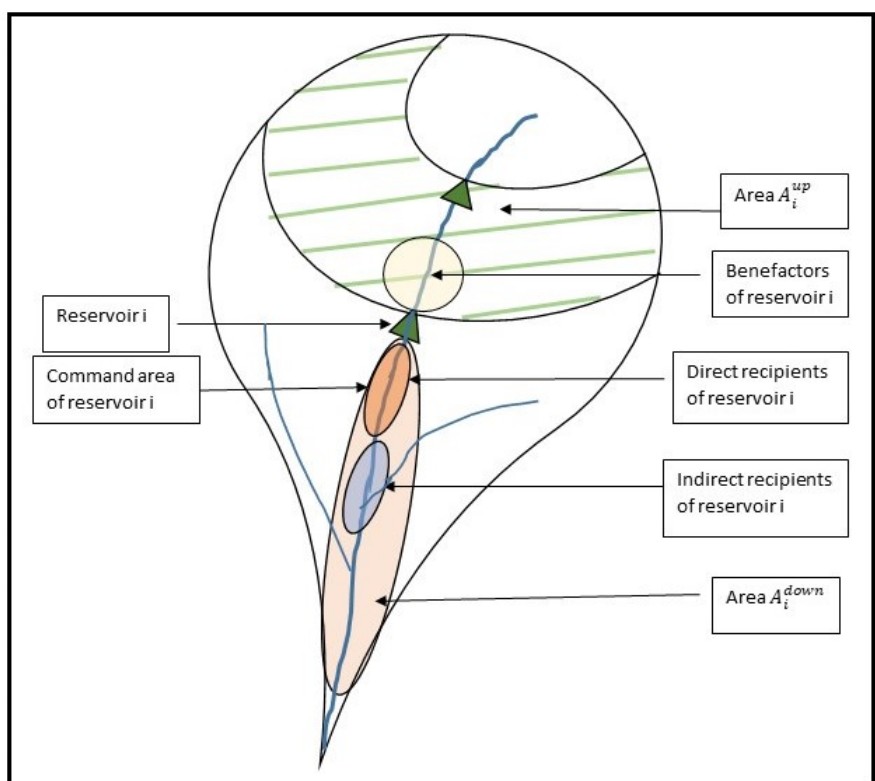

**Figure 3.** Benefactors, direct recipients and indirect recipients of a single reservoir as applicable to the estimation procedure in Equations (9)–(12).

It is assumed that for any given reservoir *i* which captures an average volume of $s_i$ in a given year, its benefactors are a fraction of the population living in that part of the basin between reservoir *i* and the reservoir immediately upstream (area $A_i^{up}$ Figure 3). It is assumed that this fraction is equivalent to the ratio between $s_i$ and the MAR of $A_i^{up}$. If the MAR of $A_i^{up}$ is $q_i^{up}$ and the population of $A_i^{up}$ is $p_i^{up}$, then the number of benefactors ($n_b$) is estimated as

$$n_b = p_i^{up} \times \left( \frac{s_i}{q_i^{up}} \right) \tag{9}$$

subject to the condition that

$$n_b \leq p_i^{up}$$

The decision to consider only a fraction of the population of area $A_i^{up}$ as benefactors was based on the consideration that reservoir *i* may not capture the total MAR generated

in area $A_i^{up}$ in a given year, especially if the reservoir is small but only a fraction of it is equivalent to $\left(\frac{s_i}{q_i^{up}}\right)$. This formulation also takes into account the impact of the size of the reservoir on the number of benefactors; i.e., the larger the reservoir, the larger the number of benefactors.

It is recognized that runoff from even further upstream than the reservoir immediately upstream of reservoir $i$ can contribute to reservoir $i$; however, to simplify the estimation procedure this runoff is not considered in Equation (9).

It is assumed that the fraction of the downstream population served by upstream runoff is always proportional to the ratio between the quantity of upstream runoff and the total incremental runoff available downstream ($q_i^{up} + q_i^{down}$). Here, $q_i^{down}$ is the MAR generated in that part of the basin which is affected by runoff held by reservoir $i$ (area $A_i^{down}$ Figure 3). Area $A_i^{down}$ is taken as the aggregated area of minor sub-basins closest to the path of the main river between reservoir $i$ and the outlet of the entire basin. This takes into account that the benefits and hydrological impacts of reservoirs can cascade downstream even up to the very end of the river basin. If the total population of the affected area is assumed as $p_i^{down}$, then the number of people who were served by the upstream runoff held by the reservoir before the reservoir was built but lose this benefit once the reservoir is established (indirect recipients $n_{ir}$) is estimated as

$$n_{ir} = p_i^{down} \times \left(\frac{s_i}{q_i^{up} + q_i^{down}}\right) \tag{10}$$

Once the reservoir is built, the exclusive benefit to those in the command area is the reservoir yield. This yield will be zero when there is no reservoir but will gradually rise with increasing reservoir capacity. Assuming that direct recipients and indirect recipients are two independent groups but part of the population of $A_i^{down}$, the number of direct recipients (exclusive beneficiaries $n_{dr}$) is estimated as

$$n_{dr} = p_i^{down} \times \left(\frac{y_i}{q_i^{up} + q_i^{down}}\right) \tag{11}$$

Estimates of $n_{ir}$ and $n_{dr}$ are made subject to the conditions that

$$n_{ir}, \ n_{dr} \leq p_i^{down}$$
$$n_{dr} + n_{ir} \leq p_i^{down}$$

Estimates of $n_b$, $n_{dr}$ and $n_{ir}$ are made in this manner independently for each reservoir. The overall NE for the basin is estimated as

$$\text{NE} = \frac{\sum_{i=1}^{n} n_b + \sum_{i=1}^{n} n_{dr} + \sum_{i=1}^{n} n_{ir}}{\sum_{i=1}^{n} n_{dr}} \tag{12}$$

where $n$ = the total number of reservoirs in the basin and subject to the conditions that $\sum_{i=1}^{n} n_b \leq$ the total population of the basin and $\sum_{i=1}^{n} n_{dr} + \sum_{i=1}^{n} n_{ir} \leq$ the total population living in the affected area of the basin from upstream to downstream.

The number of actual direct recipients may be significantly higher than these estimates since the reservoir yield is delivered to the beneficiaries in a more organized manner than natural runoff and can even be transferred upstream of the reservoir and delivered to an area extending laterally beyond $A_i^{down}$ (Figure 3). Despite assuming that the number of indirect recipients is proportional to the decrease in the downstream runoff, this number may be higher as the reservoir may impact both the quantity of flow and its timing along with numerous associated services. Similarly, while the estimates are made independently for each reservoir, there might be considerable overlap between these numbers in actual situations. However, since the main intention here is to compare differences between the

NE values of different reservoir system configurations rather than estimate absolute NE values, the simplifications are deemed reasonable. This indicator is sensitive to the storage capacity, location and spatial distribution of reservoirs and is unitless.

*2.4. Comparison of Reservoir System Configurations*

An integrated index (INT) was formulated to compare different reservoir system configurations on the basis of their performance on indicators WS Yield, EF Yield, MRRI, $MRCI_{internal}$ and NE. Indicator values across the different reservoir configurations were normalized to a common scale before combining (as described in Section 4.5) such that the most desirable value received the highest score and the most undesirable value received the lowest score. The INT value for each reservoir configuration was computed by aggregating unweighted individual indicator scores and expressing the aggregated score as a percentage of the maximum possible score. Different reservoir system configurations were subsequently compared based on this percentage score. To assess the sensitivity of INT to the number and type of indicators combined and the weights assigned to each indicator, the five indicators were recombined in four other ways to formulate alternative integrated indices (named INT1 to INT4). The results obtained by considering INT were compared with those obtained by considering the other four alternative indices. Further details of this comparison are discussed in Section 4.5.

## 3. Example Case Studies

### 3.1. Example Case Study Basins

The river basins used as examples are the Malwatu Oya and Kalu Ganga basins in the Dry and Wet Zones of Sri Lanka (Figure 4). These two basins were used because they are well known by two of the authors who reside in Sri Lanka and have been studied by other projects and input data to generate river flow time series required for the analysis were easily available. However, these approaches can be implemented in any basin for which river flow time series are available, whether observed or simulated.

The Malwatu Oya basin spans 3338 km$^2$ with an elevation range of 0 to 740 m amsl. It has a mean annual precipitation (MAP) of 1300 mm year$^{-1}$ and a natural mean annual runoff (MAR) of 0.79 km$^3$ year$^{-1}$ with a coefficient of variation of 0.47. The striking feature of this basin is its small tank cascade system (Figure 4), an irrigation system built in the third century BC, consisting of a cascade of interconnected small reservoirs storing, regulating and releasing water to paddy fields [47]. In addition to small tanks (which number 1853), this basin also has several larger irrigation reservoirs which were built simultaneously with the small tanks but later modified to suit rising water demands. The combination of small and large storage infrastructure captures about 43% of the MAR of the basin and is meant to meet irrigation and domestic needs, particularly during the dry season from May to September. However, rising water demand and recurrent droughts have resulted in water scarcity in the basin [48].

The Kalu Ganga basin is 2296 km$^2$ with an elevation range of 0 to 2250 m amsl and a MAP of 4000 mm year$^{-1}$. Compared to Malwatu Oya, Kalu Ganga produces higher and less variable natural runoff with a MAR of 8.45 km$^3$ year$^{-1}$ and a coefficient of variation of 0.14. Perennial rainfed crops such as tea and rubber dominate agriculture in Kalu Ganga (as opposed to irrigated paddy in Malwatu Oya). Flooding is an annual phenomenon in the low-lying areas of the basin [49]. At present, there is only one constructed reservoir within the basin (Kukuleganga, a run-of-the-river hydropower reservoir which stores only about 0.02% of the MAR). However, more reservoirs are envisaged to be constructed in the future to harness the river's ample water resources for interbasin transfer and hydropower [50].

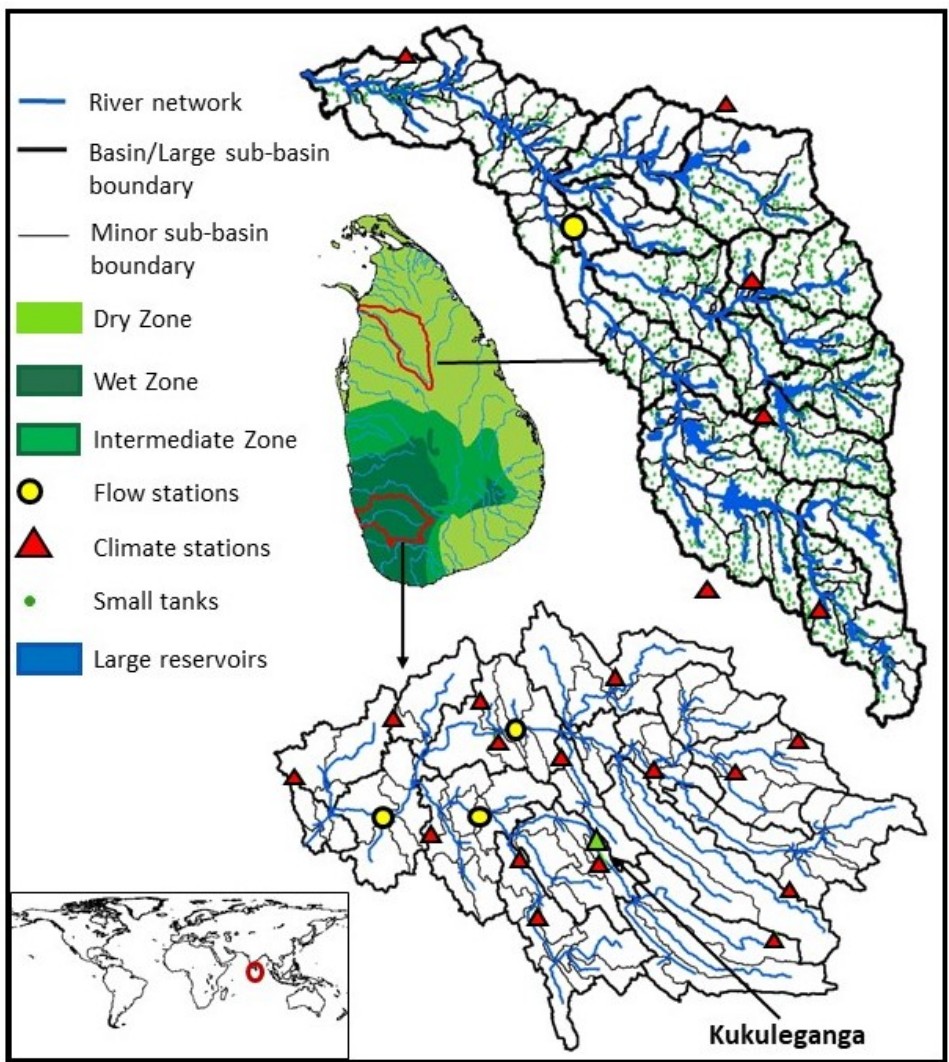

**Figure 4.** Location of the case study basins in Sri Lanka (**center**), the sub-basins, stream networks and climate stations within the Malwatu Oya (**top**) and the Kalu Ganga (**bottom**) basins (source: Eriyagama et al. [20]).

*3.2. Data and Reservoir System Configuration Scenarios*

All analyses presented in this paper were performed using time series of simulated natural monthly flows for the period 1961–2013 at the outlets of 148 and 114 minor sub-basins of the Malwatu Oya and Kalu Ganga basins (Figure 4). The same dataset was used for the analysis in [20]. Natural or unregulated flows were simulated by using the Soil and Water Assessment Tool (SWAT) model using input climate data at the locations shown in Figure 4. The calibration and validation of the two models, described in detail by Eriyagama et al. [20], are not repeated in this paper. After calibration, the output flow records were naturalized by removing all existing reservoirs and water management practices and re-executing the models.

The previous analysis [20] used three reservoir system configurations designed such that for a given storage capacity, the storage infrastructure in the basin progressed from one large reservoir at the outlet of the entire basin to several small reservoirs distributed in sub-basins. The sub-basin delineation followed that of the Department of Agrarian Development. The same reservoir system configurations (Configurations 1, 2 and 3), were used for the current analysis (Figures 5 and 6). The capacity of individual reservoirs decreases gradually from Configuration 1 to Configuration 3 and was designed to distribute the cumulative storage capacity of the entire basin according to the ratio between the MAR

of sub-basins draining into each reservoir since such a distribution was found to maximize cumulative yields [20]. Table 1 shows the main features of each configuration. Different reservoir arrangements based on Configurations 1, 2 and 3 were formulated to investigate the response of indicators to changes in the cumulative storage capacity, number, location and spatial distribution of reservoirs within the basins as summarized in Table 2. When formulating the reservoir arrangements, the cumulative storage capacity in each basin was made to vary from 0.05 to 1.0 MAR units for illustrative purposes. Actual cumulative storage capacities in river basins may extend well beyond this range.

**Table 1.** Main features of the reservoir system Configurations 1, 2 and 3.

| Basin | Malwatu Oya | | | Kalu Ganga | | |
|---|---|---|---|---|---|---|
| Configuration | 1 | 2 | 3 | 1 | 2 | 3 |
| Number of reservoirs | 1 | 15 | 148 | 1 | 17 | 114 |
| Average basin area draining into a single reservoir (km$^2$) | 3338 | 223 | 22.6 | 2296 | 172 | 25.7 |
| Average MAR draining into a single reservoir (km$^3$ year$^{-1}$) | 0.79 | 0.06 | 0.006 | 8.45 | 0.51 | 0.075 |

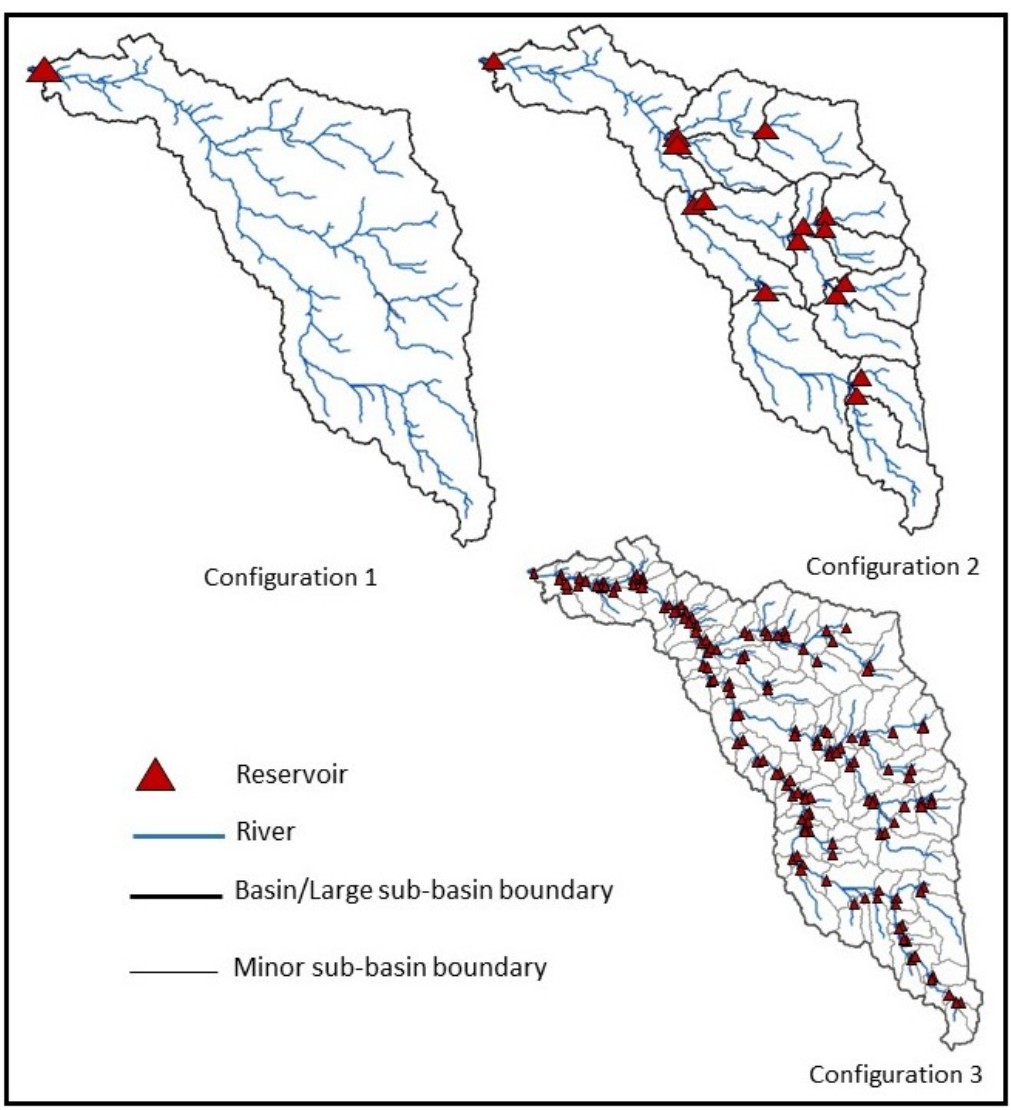

**Figure 5.** Reservoir system configurations in Malwatu Oya. Configurations range from 1 large reservoir at the river outlet to 15 medium-size reservoirs in large sub-basins and 148 smaller reservoirs in minor sub-basins (source: Eriyagama et al. [20]).

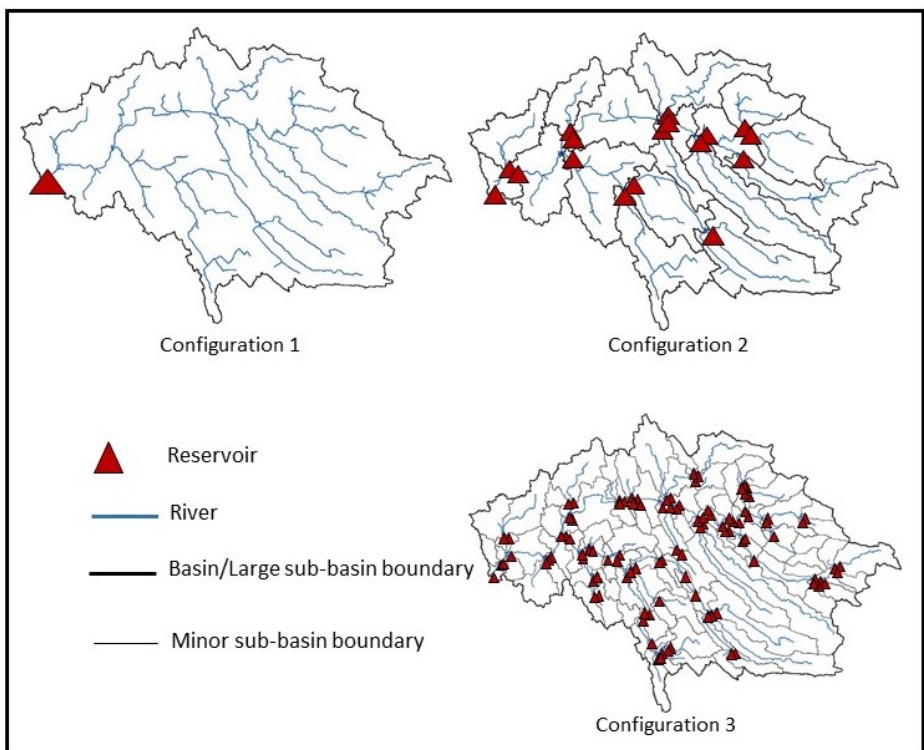

**Figure 6.** Reservoir system configurations in Kalu Ganga. Configurations range from 1 large reservoir at the river outlet to 17 medium-size reservoirs in large sub-basins and 114 smaller reservoirs in minor sub-basins [20].

**Table 2.** Reservoir arrangements tested for each indicator.

| Indicator | Cumulative Storage Capacity Range in Basin (MAR units) | Reservoir Arrangement | Focus of Investigation |
|---|---|---|---|
| WS Yield and EF Yield | 0.05–1.0 | Reservoirs are placed at all locations in Configurations 1, 2 and 3 | The impact of varying levels of cumulative storage capacity and spatial distribution of reservoirs |
| MRRI | 0.05–1.0 | Only a single reservoir is placed in the entire basin at any given time. The location of the reservoir is changed from upstream to downstream so that it is at the outlet of each minor sub-basin of Configuration 3, starting with the first-order rivers of the most upstream sub-basin in Configuration 2. Reservoirs are progressively added one by one from upstream to downstream under Configurations 1, 2 and 3, starting with first-order rivers. | The impact of the location of reservoirs and the spatial distribution of the cumulative storage capacity of the basin |
| $MRCI_{internal}$ and $MRCI_{external}$ | Storage capacity does not influence $MRCI_{internal}$ and $MRCI_{external}$ | As above | As above |
| NE | 0.05–1.0 | Reservoirs are placed at all locations in Configurations 1, 2 and 3 | Differences in social impacts between centralized large and distributed small reservoirs under varying cumulative storage capacities |

## 4. Results and Discussion

### 4.1. WS Yield and EF Yield

The cumulative WS Yields from the entire basin under Configurations 1, 2 and 3 for gradually increasing storage capacities are shown in Table 3. It is evident that the cumulative WS Yield from a network of reservoirs declines with increasing levels of spatial distribution even though the aggregated storage capacity remains the same; i.e., the WS Yield is highest under Configuration 1 and gradually decreases when moving to Configurations 2 and 3. However, as observed by [13], with increasing levels of spatial distribution, the spatially unequal distribution of access to the economic benefits of reservoir water decreases, increasing the number of people who benefit from the water yield. This aspect is further discussed in Section 4.4 under normalized externality. Table 3 also shows the EF Yield corresponding to each WS Yield. As expected, for each cumulative storage capacity, the EF Yield increases when moving from Configuration 1 to Configuration 3.

**Table 3.** Cumulative WS and EF Yields from the entire basin under Configurations 1, 2 and 3 for gradually increasing cumulative storage capacities.

| Storage Capacity (MAR Units) | WS Yield (MAR Units) | | | | | | EF Yield (MAR Units) | | | | | |
|---|---|---|---|---|---|---|---|---|---|---|---|---|
| | Malwatu Oya | | | Kalu Ganga | | | Malwatu Oya | | | Kalu Ganga | | |
| | 1 | 2 | 3 | 1 | 2 | 3 | 1 | 2 | 3 | 1 | 2 | 3 |
| 0.05 | 0.16 | 0.11 | 0.10 | 0.27 | 0.22 | 0.18 | 0.84 | 0.89 | 0.90 | 0.73 | 0.78 | 0.82 |
| 0.1 | 0.20 | 0.16 | 0.15 | 0.43 | 0.35 | 0.29 | 0.80 | 0.84 | 0.85 | 0.57 | 0.65 | 0.71 |
| 0.4 | 0.37 | 0.35 | 0.33 | 0.87 | 0.72 | 0.63 | 0.63 | 0.65 | 0.67 | 0.13 | 0.28 | 0.37 |
| 0.8 | 0.53 | 0.50 | 0.48 | 0.94 | 0.84 | 0.76 | 0.47 | 0.50 | 0.52 | 0.06 | 0.16 | 0.24 |
| 1 | 0.59 | 0.56 | 0.53 | 0.96 | 0.88 | 0.80 | 0.41 | 0.44 | 0.47 | 0.04 | 0.12 | 0.20 |

Note: Storage capacities and yields are presented in MAR units to enable comparisons.

### 4.2. Flow Regulation

Figure 7 shows the variation of MRRI with the normalized upstream basin area impounded when the location of a single reservoir of a given capacity is moved from upstream to downstream (arrangement (a) in Table 2). Each line in Figure 7 corresponds to a unique storage capacity of a single reservoir. It shows that the overall MRRI gradually increases as the reservoir location changes from upstream to downstream (as the upstream area impounded increases), although the capacity of the reservoir remains constant. The MRRI becomes equal to the DOR only if the reservoir is located at the most downstream end of the river. Figure 7 illustrates that the MRRI is able to capture the volume of flow regulation and the impact of the location of the reservoir due to the distance factor.

Figure 8 shows the final MRRI for Configurations 1, 2 and 3 (arrangement (b) in Table 2). In all three configurations, the final MRRI increases as the cumulative storage capacity in the basin becomes larger. However, this increase is more gradual in Configurations 2 and 3 due to the effect of the distribution factor. The figure also illustrates that a small number of large reservoirs (Configurations 1 and 2) results in a higher overall MRRI than a large number of small reservoirs (Configuration 3) even if the aggregated storage capacity is the same. The MRRI is equal to the DOR only under Configuration 1. Under Configurations 2 and 3, the DOR is modified by distribution factors 0.283 and 0.093 in Malwatu Oya and 0.207 and 0.090 in Kalu Ganga. The MRRI is sensitive to the annual quantity of flow stored by reservoirs and its spatial distribution and indirectly to the number of reservoirs in the river network due to the effect of the distance and distribution factors.

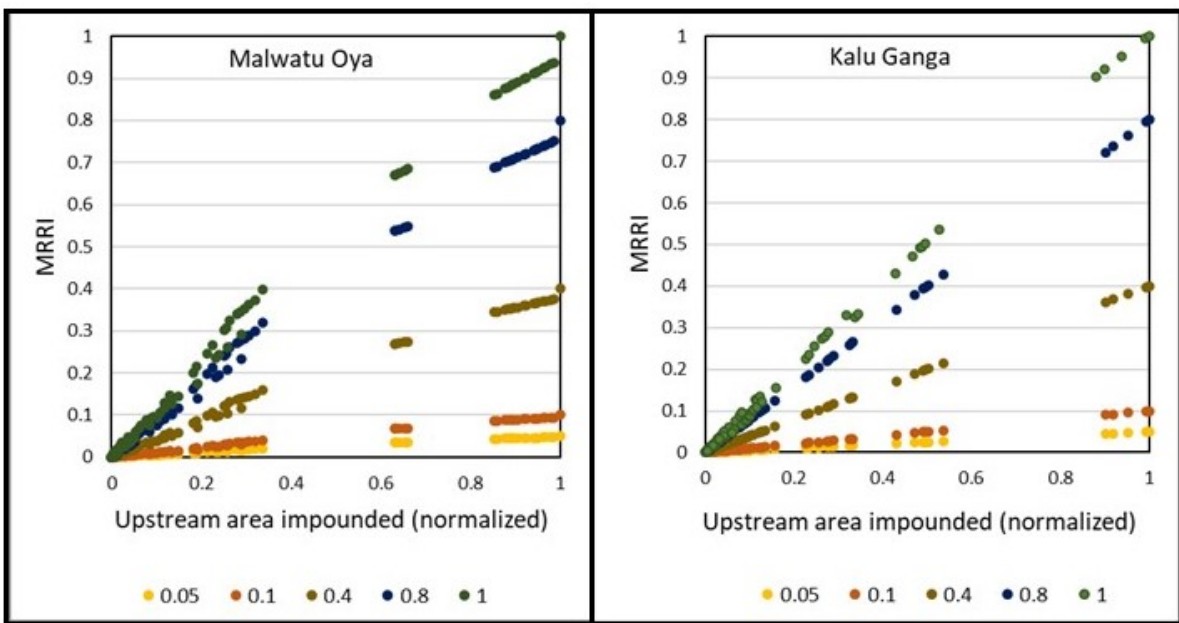

**Figure 7.** Variation of modified river regulation index (MRRI) with normalized upstream basin area impounded when a single reservoir of capacity 0.05 to 1 MAR (of the entire river) is placed at the outlet of each minor sub-basin identified in Configuration 3 of Malwatu Oya and Kalu Ganga basins.

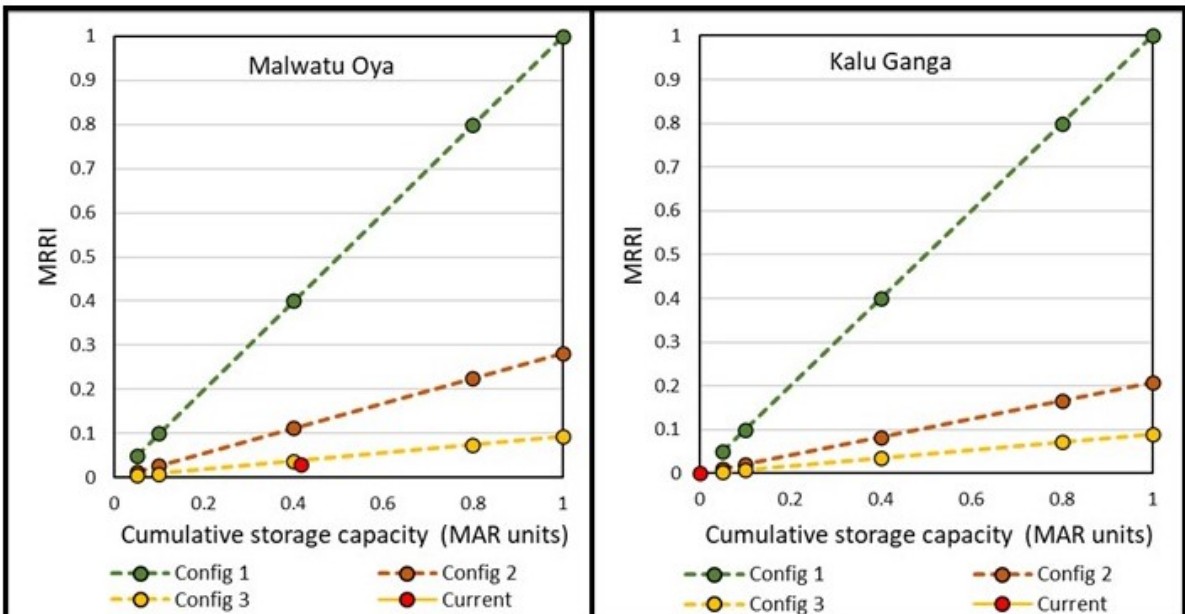

**Figure 8.** The final modified river regulation index (MRRI) for five aggregated storage capacities ranging from 0.05 to 1 MAR under Configurations 1, 2 and 3 in Malwatu Oya and Kalu Ganga basins.

Figure 8 (left) also shows that the MRRI for the existing reservoir system configuration in Malwatu Oya is extremely low. Although the basin stores 0.43 of the MAR and has a DOR of 0.43, its MRRI is 0.029 due to the high level of distribution of reservoirs in the landscape (distribution factor 0.071). As there is only a single reservoir in Kalu Ganga storing 0.02% of the MAR of the basin, its current MRRI is extremely small (Figure 8 right).

It is generally understood that the scope and scale of negative impacts due to regulation by a cluster of small reservoirs distributed in the landscape are lower than those of a single lumped reservoir with the same aggregated capacity (e.g., [13]). The MRRI aims to reflect this understanding, although the values of MRRI are not intended to be proportional to

the actual physical impacts on the ground. Although the MRRI was formulated to serve the purpose of this research, it can also be used in other situations to compare reservoir configurations having similar storage capacities but different spatial distributions.

### 4.3. River Connectivity

Figure 9 shows the variation of $MRCI_{internal}$ and $MRCI_{external}$ with the normalized upstream basin area impounded when the location of a single reservoir is changed from upstream to downstream (arrangement (a) in Table 2). It illustrates that when a new reservoir is introduced to a first-order stream, both indices remain relatively high (nearly 100) since only a small segment of the network becomes disconnected. The lowest internal connectivity (approximately 40%) is observed when a new reservoir is placed on the main stem of a river so the normalized upstream basin area impounded is approximately 0.53. When a reservoir is placed downstream of this point on the main stem of the river, the internal connectivity index again increases until it reaches 100 when a reservoir is placed at the outlet to the sea. The $MRCI_{external}$ linearly decreases as the reservoir location is moved from upstream to downstream, and its value reaches zero when a reservoir is placed at the outlet to the sea.

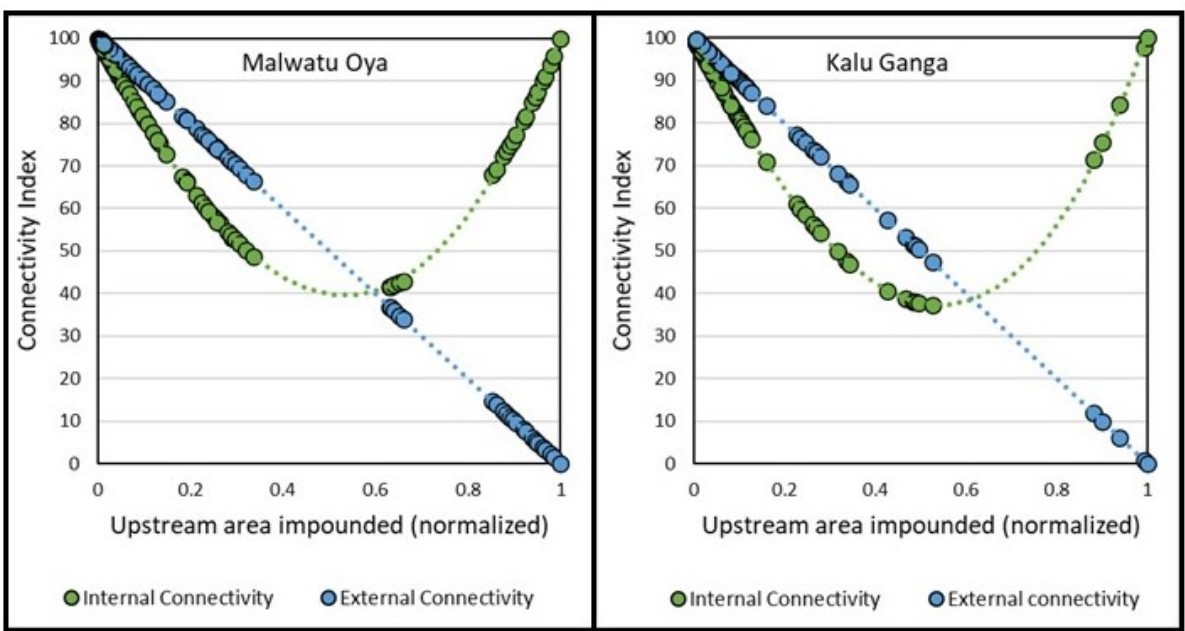

**Figure 9.** Variation of the internal and external river connectivity indices with normalized upstream basin area impounded when a single reservoir is placed in the basin and its location is moved from upstream to downstream to the outlet of each minor sub-basin identified in Configuration 3 for Malwatu Oya and Kalu Ganga basins.

If the internal and external connectivity indices are combined into a composite index by taking their mean, this combined index has its lowest value on the main stem of the river when the normalized upstream impounded area is approximately 0.72. Therefore, considering both $MRCI_{internal}$ and the combined index, the lowest internal and external connectivity is generated if a new reservoir is placed on the main stem of a previously unregulated river between the two locations where the normalized upstream basin area is approximately 0.5 and 0.75.

Figure 10 compares the graphs of $MRCI_{internal}$ and $MRCI_{external}$ with those of $DCI_{internal}$ and $DCI_{external}$ of Cote et al. [44] (Equations (4) and (5)) for the Malwatu Oya basin. It shows there is a minimal discrepancy between the graphs of $MRCI_{external}$ and $DCI_{external}$. However, the weighting introduced in Equation (6) results in enhancing the contrast between internal connectivity estimates of barriers placed on upstream reaches and main

stems of a river and a slight downstream migration of the location at which internal connectivity reaches a minimum. This location for $DCI_{internal}$ is where the total length of the entire river is bisected (where the normalized upstream basin area is approximately equal to 0.5). In the case of DCI, the 0.5–0.75 range in normalized upstream basin area remains where the lowest internal and external connectivity is generated by a new reservoir introduced to a previously unregulated river. Therefore, the results for DCI and MRCI are largely compatible, while MRCI is better at emphasizing the differences in internal river connectivity produced by different sites of a new reservoir.

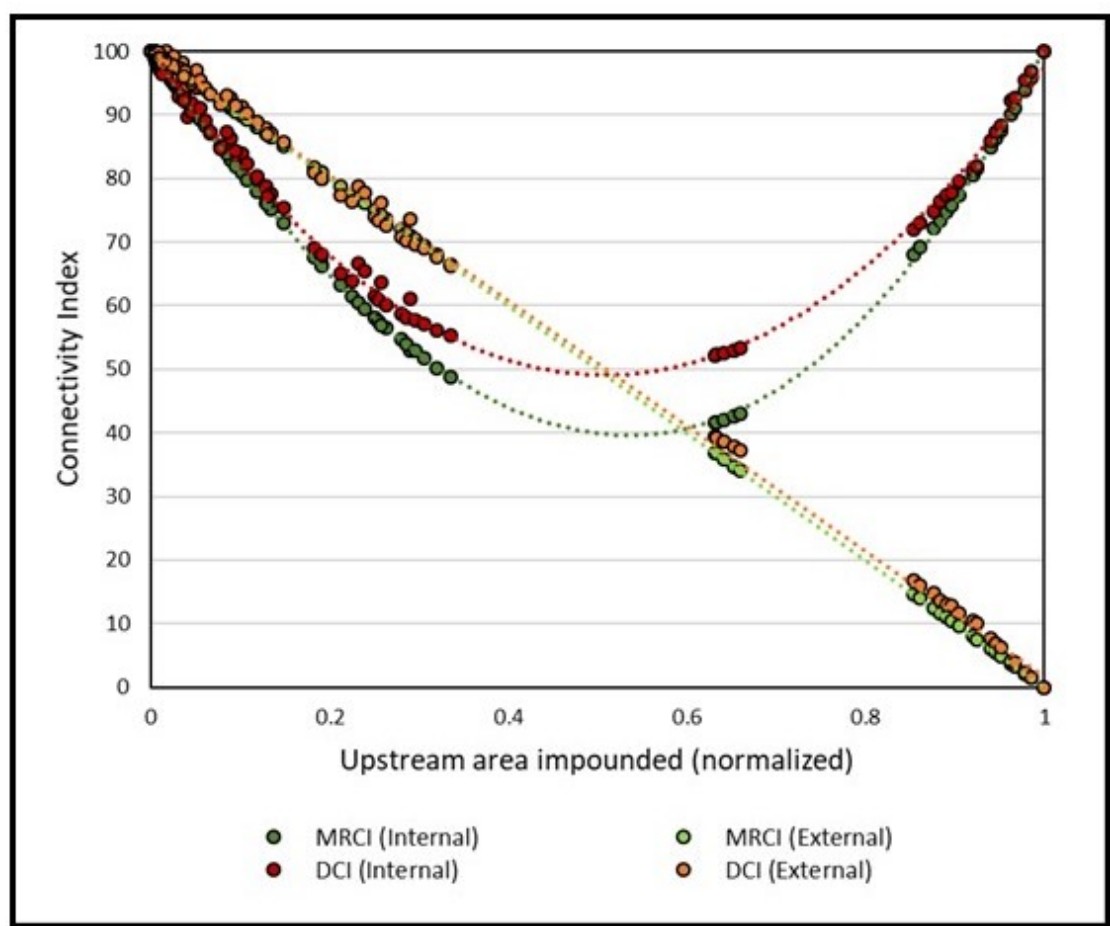

**Figure 10.** Comparison of the variation of MRCI and DCI for the Malwatu Oya basin when a single reservoir is placed in the basin and its location is moved from upstream to downstream to the outlet of each minor sub-basin identified in Configuration 3.

The behavior of $MRCI_{internal}$ with progressive addition of reservoirs from upstream to downstream (arrangement (b) in Table 2) is shown in Figure 11. Figure 11 (top) illustrates that the impact on $MRCI_{internal}$ is higher as the first few reservoirs are added to the network but this impact diminishes as the number of reservoirs increases (the curve becomes asymptotic to the horizontal axis). It also shows that the relationship between $MRCI_{internal}$ and the upstream basin area impounded follows a similar path irrespective of whether reservoirs are added at a large sub-basin level (Configuration 2) or a minor sub-basin level (Configuration 3). This means that the addition of further reservoirs in the upstream reaches of an already disconnected network has a negligible influence on the overall internal connectivity of the river network. Even if new reservoirs are added to the main stems of an already disconnected network, the overall connectivity decreases only slightly.

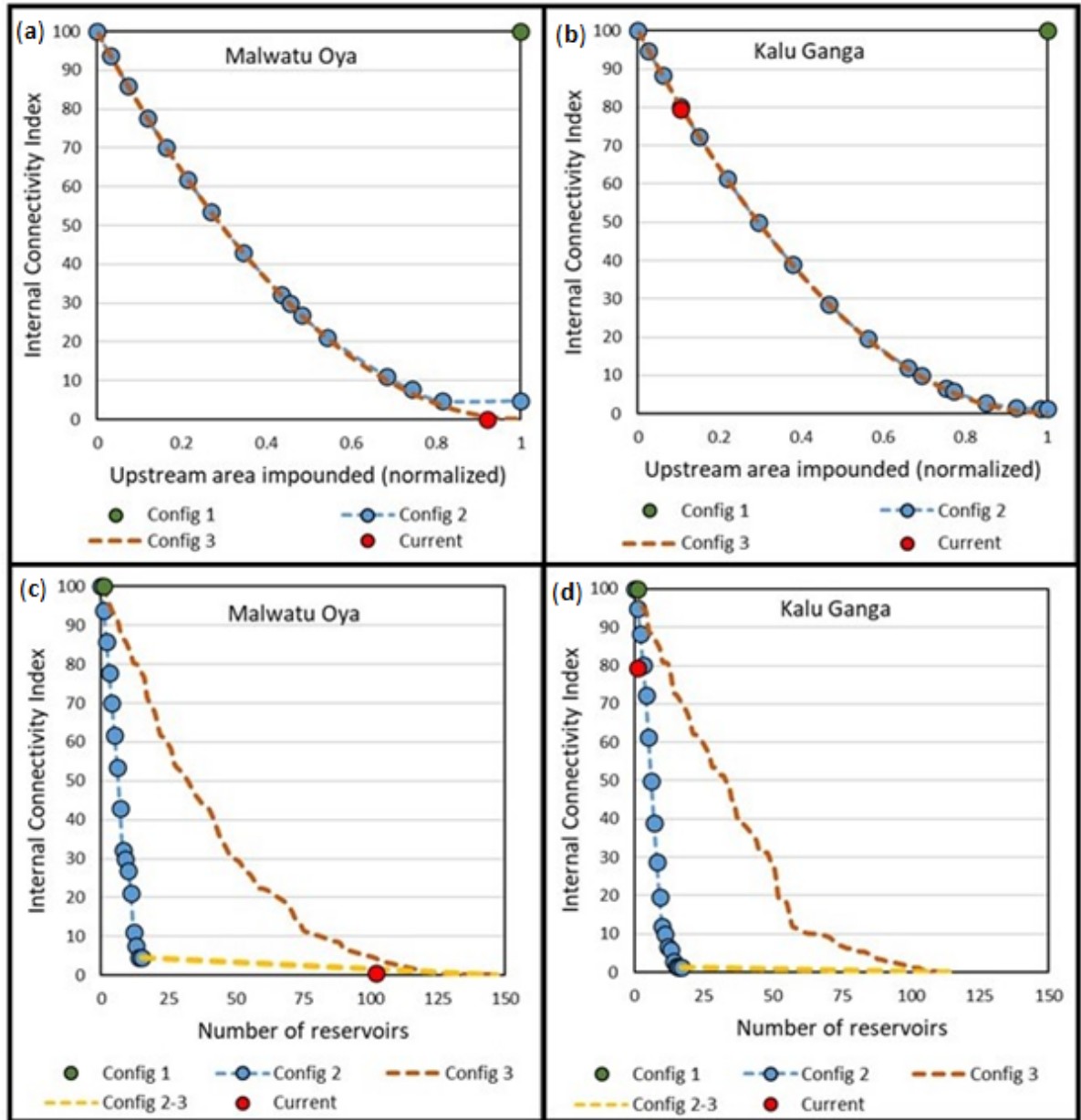

**Figure 11.** Variation of the internal river connectivity index with normalized upstream basin area impounded (**a**,**b**) and the number of reservoirs (**c**,**d**) for Malwatu Oya and Kalu Ganga basins when reservoirs are progressively added from upstream to downstream under Configurations 1, 2 and 3.

This observation is further illustrated in Figure 11 (bottom), which shows that $MRCI_{internal}$ decreases rapidly when reservoirs are added at a large sub-basin level (Configuration 2) but decreases at a much lower rate when additional reservoirs are added at a minor sub-basin level (Configuration 2 and 3) within those large sub-basins. On the other hand, if a similar number of reservoirs are introduced at a minor sub-basin level (Configuration 3), $MRCI_{internal}$ shows only a gradual decrease. Figure 11 also illustrates the current levels of internal connectivity of Malwatu Oya and Kalu Ganga. The former's is extremely low due to the presence of a large number of reservoirs whereas the latter is as high as 80% since it only has a single reservoir.

$MRCI_{external}$ decreases linearly as reservoirs are cumulatively added from upstream to downstream and becomes zero if a reservoir is placed at the sea outlet. Since it depends only on the basin area directly connected with the sea, it remains unchanged even if new reservoirs are added upstream of the furthest downstream reservoir and hence it is not

illustrated here. While the current external connectivity of Kalu Ganga is nearly 90%, it is nearly zero in Malwatu Oya.

The behavior of the alternative connectivity indices formulated in this research is similar to those of Cote et al. [44] and Grill [19], although those indices are based on different sets of variables. Although the alternative indices were formulated specifically for this research, they can be adapted to measure the degree of connectivity in river networks and to compare different reservoir system configurations in other river basins when explicit information on river lengths or volumes is not available.

### 4.4. Normalized Externality

Figure 12 shows the behavior of NE for the five instances of aggregated storage capacities ranging from 0.05 to 1 MAR for both basins under reservoir Configurations 2 and 3. Configuration 1 is omitted from Figure 12 since in this case the estimated NE is extremely large (near infinity), considering there are only a small number of direct beneficiaries and indirect recipients further downstream of the reservoir. The figure shows that the NE increases with increasing accumulated storage capacity in river basins. This confirms that, in general, the ratio of the total number affected to the total number of beneficiaries increases as the basin storage capacity expands. However, at any given storage capacity, the NE value (i.e., the distribution of access to the benefits of stored water) differs between reservoir system configurations. The graphs for Malwatu Oya (Figure 12 left) illustrate that a higher equitable distribution of water access and a smaller number of affected people (lower NE values) results from a large number of small reservoirs distributed in the landscape (Configuration 3) rather than from a small number of concentrated large reservoirs of the same aggregated storage capacity (Configuration 2). However, this distinction is less apparent in Kalu Ganga (Figure 12 right). Even in Malwatu Oya, this difference disappears at very small aggregated storage capacities such as 0.05 MAR units and very large aggregated storage capacities such as 1 MAR unit.

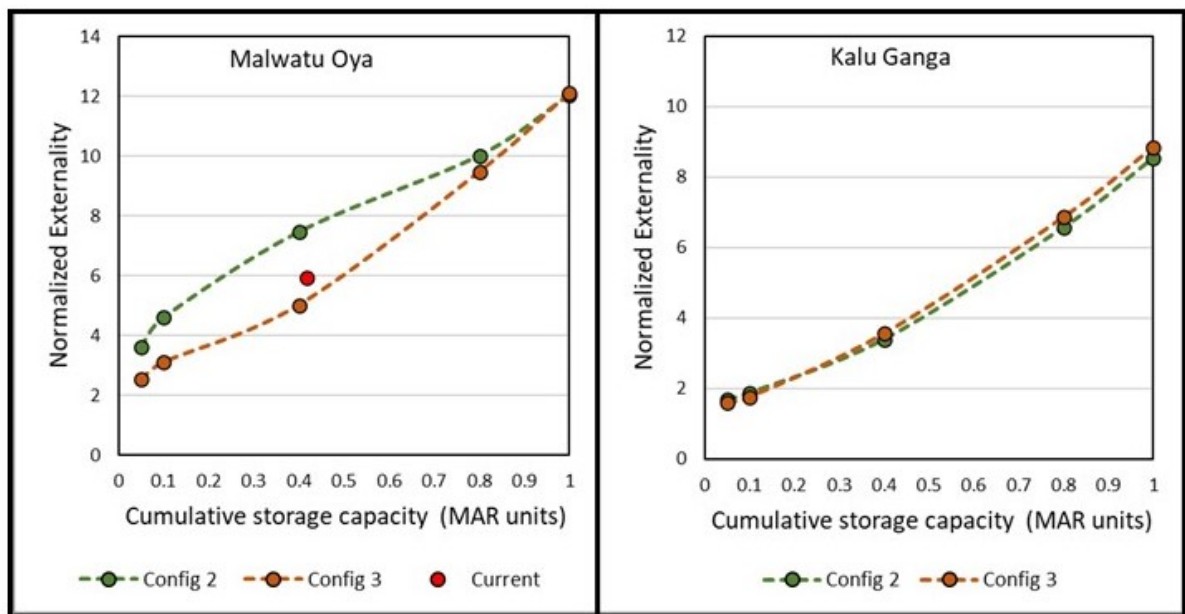

**Figure 12.** Final normalized externality (NE) for five aggregated storage capacities ranging from 0.05 to 1 MAR under Configurations 2 and 3 in Malwatu Oya and Kalu Ganga basins.

Absolute NE values for all configurations are lower in Kalu Ganga than in Malwatu Oya (Table 4). For instance, if a 0.1 MAR cumulative storage capacity is distributed in Malwatu Oya under Configuration 2, for every beneficiary there are four other affected actors (since NE = 5), whereas under Configuration 3, there are only two other affected actors. However, in the case of Kalu Ganga, there is only one other affected actor except

for the beneficiary (since NE = 2) under both Configurations 2 and 3. The main reason for the differences in the behavior of NE between the two river basins is the difference in their shapes. Any upstream occurrence in Malwatu Oya has a higher potential to influence its downstream area due to its elongated shape and higher bifurcation ratio (length 164 km and bifurcation ratio 5.25; [51]). In Kalu Ganga, this potential is lower due to its low elongation and low bifurcation ratio (length 120 km and bifurcation ratio 3.85; [51]). Therefore, the geometry of a river basin also plays a major role in cascading impacts downstream. The current NE of Malwatu Oya (Figure 12) is estimated as 6. However, its actual value is likely to be significantly smaller, considering the highly distributed nature of small reservoirs in the basin. The current NE of Kalu Ganga was not estimated since the only existing reservoir is a run-of-river hydropower reservoir and the approximations explained earlier are not directly applicable in this case.

**Table 4.** Overall NE values for Malwatu Oya and Kalu Ganga basins under Configurations 2 and 3 (rounded off to the nearest integer).

| Basin | Malwatu Oya | | Kalu Ganga | |
|---|---|---|---|---|
| Configuration | 2 | 3 | 2 | 3 |
| Number of reservoirs | 15 | 148 | 17 | 114 |
| Cumulative storage capacity in Basin (MAR units) | Overall NE Value | | | |
| 0.05 | 4 | 2 | 2 | 2 |
| 0.1 | 5 | 3 | 2 | 2 |
| 0.4 | 7 | 5 | 3 | 3 |
| 0.8 | 10 | 9 | 7 | 7 |
| 1 | 12 | 12 | 9 | 9 |

The approximations used in this research to estimate NE, although not having explicit data on affected persons, produce results that reasonably agree with those anticipated by the original authors of the index [13]. However, due to the assumptions and approximations applied, they are more suitable for use in comparing different configurations and different river basins rather than deriving absolute numbers of NE.

*4.5. Comparison of Reservoir System Configurations*

Table 5 shows the individual indicator values for reservoir system Configurations 1, 2 and 3 for the cumulative storage capacity range 0.05 to 1 MAR units. Table 5 also includes indicator values for the existing reservoir configuration in Malwatu Oya (storage capacity 0.43 MAR units). Since the current NE was not estimated for Kalu Ganga, an integrated index value (INT) was not calculated for the existing conditions in Kalu Ganga.

Table 5 illustrates WS Yield, EF Yield, MRRI, $MRCI_{internal}$ and NE values for a range of alternative storage scenarios in the Malwatu Oya and Kalu Ganga basins (16 for Malwatu Oya and 15 for Kalu Ganga). The indicator values in Table 5 for each alternative storage scenario were given a score (of 1–16 for Malwatu Oya and 1–15 for Kalu Ganga) in such a way that the most desirable value received the highest score, while the most undesirable value received the lowest. Accordingly, the highest WS Yield, EF Yield and $MRCI_{internal}$ values and the lowest MRRI and NE values received the highest scores. A final score for each alternative storage scenario was computed by aggregating the scores for individual indicators. The aggregated scores (out of the maximum possible) were expressed on a percentage scale to compute the integrated index (INT) as shown in Table 5. Figure 13 shows the variation of INT across the different storage scenarios for the two study basins.

**Table 5.** Values for WS Yield, EF Yield, MRRI, *MRCI*<sub>internal</sub> and NE under different reservoir system configurations and cumulative reservoir capacities for Malwatu Oya and Kalu Ganga basins.

| Config. | Storage Scenario | Storage Capacity (MAR Units) | Storage Capacity (Million m$^3$) | Maximum WS Yield (Million m$^3$ year$^{-1}$) | EF Yield (Million m$^3$ year$^{-1}$) | MRRI (years) | MRCIinternal | NE | INT |
|---|---|---|---|---|---|---|---|---|---|
| | | | | **Malwatu Oya** | | | | | |
| 1 | 1 | 0.05 | 39.48 | 125.56 | 664.13 | 0.05 | 100 | VL | 56 |
| | 2 | 0.1 | 78.97 | 159.52 | 630.18 | 0.1 | 100 | VL | 52 |
| | 3 | 0.4 | 315.88 | 294.56 | 495.14 | 0.4 | 100 | VL | 47 |
| | 4 | 0.8 | 631.76 | 420.12 | 369.58 | 0.8 | 100 | VL | 46 |
| | 5 | 1 | 789.70 | 469.08 | 320.62 | 1 | 100 | VL | 44 |
| 2 | 6 | 0.05 | 39.48 | 88.57 | 701.13 | 0.014 | 4.669 | 4 | 68 |
| | 7 | 0.1 | 78.97 | 126.47 | 663.22 | 0.028 | 4.669 | 5 | 65 |
| | 8 | 0.4 | 315.88 | 276.40 | 513.29 | 0.113 | 4.669 | 7 | 54 |
| | 9 | 0.8 | 631.76 | 397.69 | 392.00 | 0.226 | 4.669 | 10 | 49 |
| | 10 | 1 | 789.70 | 438.38 | 351.32 | 0.283 | 4.669 | 12 | 45 |
| 3 | 11 | 0.05 | 39.48 | 78.58 | 711.12 | 0.005 | 0.100 | 3 | 68 |
| | 12 | 0.1 | 78.97 | 115.56 | 674.14 | 0.009 | 0.100 | 3 | 65 |
| | 13 | 0.4 | 315.88 | 259.56 | 530.14 | 0.037 | 0.100 | 5 | 55 |
| | 14 | 0.43 | 339.57 | 244.02 | 353.49 | 0.029 | 0.482 | 6 | 49 |
| | 15 | 0.8 | 631.76 | 376.46 | 413.24 | 0.075 | 0.100 | 9 | 49 |
| | 16 | 1 | 789.70 | 418.36 | 371.34 | 0.093 | 0.100 | 12 | 42 |
| | | | | **Kalu Ganga** | | | | | |
| 1 | 1 | 0.05 | 422.64 | 2240.00 | 6212.82 | 0.05 | 100 | VL | 56 |
| | 2 | 0.1 | 845.28 | 3651.62 | 4801.20 | 0.1 | 100 | VL | 51 |
| | 3 | 0.4 | 3381.13 | 7311.68 | 1141.13 | 0.4 | 100 | VL | 47 |
| | 4 | 0.8 | 6762.25 | 7937.19 | 515.62 | 0.8 | 100 | VL | 46 |
| | 5 | 1 | 8452.81 | 8038.63 | 414.19 | 1 | 100 | VL | 44 |
| 2 | 6 | 0.05 | 422.64 | 1869.66 | 6583.15 | 0.010 | 1.366 | 2 | 70 |
| | 7 | 0.1 | 845.28 | 2955.88 | 5496.93 | 0.021 | 1.366 | 2 | 65 |
| | 8 | 0.4 | 3381.13 | 6103.02 | 2349.79 | 0.083 | 1.366 | 3 | 58 |
| | 9 | 0.8 | 6762.25 | 7105.63 | 1347.19 | 0.166 | 1.366 | 7 | 50 |
| | 10 | 1 | 8452.81 | 7331.09 | 1121.73 | 0.207 | 1.366 | 9 | 45 |
| 3 | 11 | 0.05 | 422.64 | 1518.21 | 6934.60 | 0.004 | 0.064 | 2 | 68 |
| | 12 | 0.1 | 845.28 | 2409.36 | 6043.46 | 0.009 | 0.064 | 2 | 63 |
| | 13 | 0.4 | 3381.13 | 5329.01 | 3123.81 | 0.036 | 0.064 | 3 | 54 |
| | 14 | 0.8 | 6762.25 | 6429.12 | 2023.70 | 0.072 | 0.064 | 7 | 47 |
| | 15 | 1 | 8452.81 | 6763.86 | 1688.96 | 0.090 | 0.064 | 9 | 41 |

Note: VL = very large.

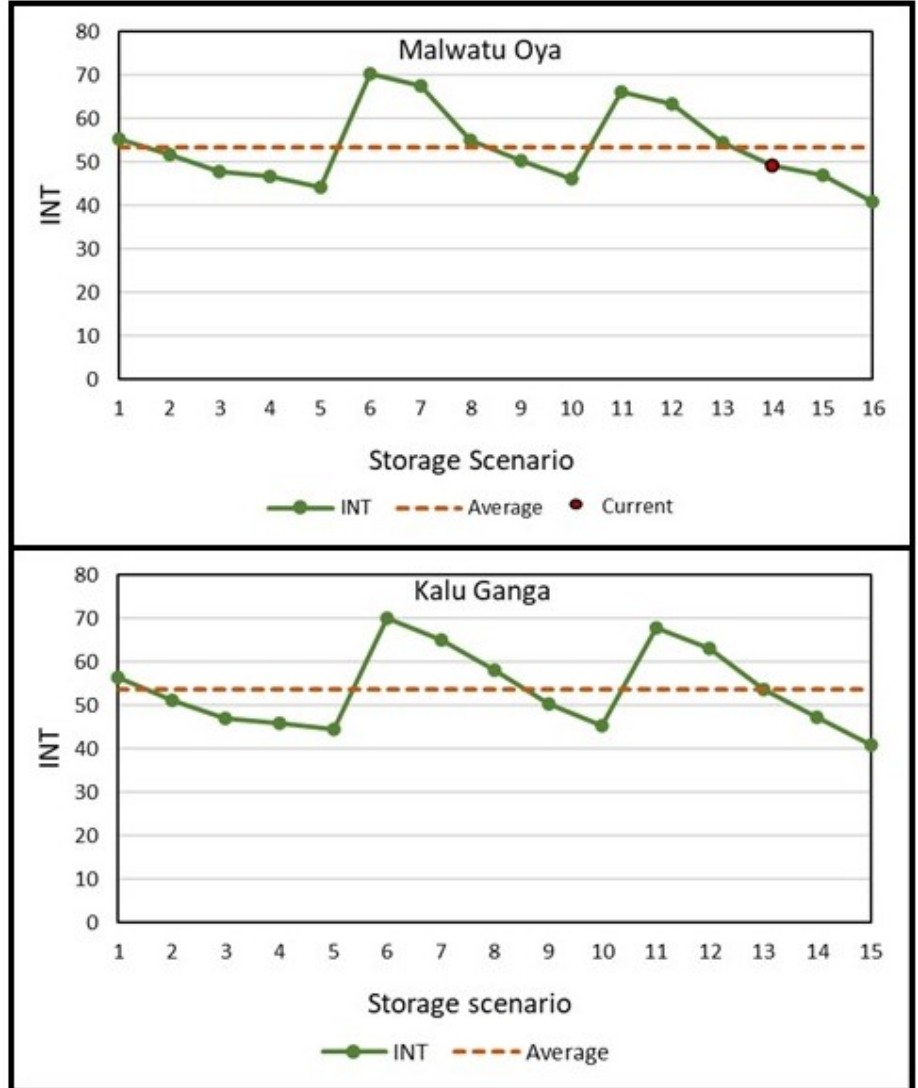

**Figure 13.** Variation of the integrated index (INT) scores across alternative storage scenarios for the Malwatu Oya and Kalu Ganga basins.

According to Figure 13, lower cumulative storage capacities produce higher INT values (e.g., storage scenarios 1, 6 and 11 where the cumulative storage capacity is 0.05 MAR units). Higher cumulative storage capacities produce lower INT values (e.g., storage scenarios 5, 10 and 16 in Malwatu Oya where the cumulative storage capacity is 1 MAR unit) under all reservoir system configurations. The average INT value for both rivers is 54. For Configuration 1 in both rivers, the INT score is above the average only if the cumulative storage capacity is as low as 0.05 MAR units, while for all other cumulative storage capacities it is below the average. For Configurations 2 and 3, the INT score is above or equal to the average up to 0.4 MAR units for both rivers. This illustrates that even larger cumulative storage capacities score high on the integrated index when reservoirs are distributed in the river network rather than being lumped. The highest INT values in both rivers are exhibited by storage scenarios 6 and 11, where a cumulative storage capacity of 0.05 MAR units is distributed under Configurations 2 and 3. The existing distributed storage configuration in Malwatu Oya falls just below the average with an INT of 49.

Figure 14 shows the variation of INT with storage capacity for the three configurations in each basin. In Malwatu Oya, two distinct cumulative storage capacity zones can be identified (Figure 14 left). The two zones are (i) cumulative storage capacity below 0.8 MAR units and (ii) cumulative storage capacity above 0.8 MAR units. When the cumulative

storage capacity in the basin is below 0.8 MAR units, both storage Configurations 2 and 3 perform equally well and better than Configuration 1. When the cumulative storage capacity in the basin is above 0.8 MAR units, the performance of all three configurations is similarly low. At 1 MAR unit storage capacity, the INT of Configuration 3 has the lowest value; i.e., the cumulative effects of a large number of small reservoirs can be as significant as those by a small number of large reservoirs of the same aggregated storage capacity, especially if the aggregated storage capacity is as large as the MAR of the river. The current reservoir configuration in Malwatu Oya lies between the two extreme curves in Figure 14.

In Kalu Ganga, Configurations 2 and 3 perform better than Configuration 1 up to a storage capacity of 0.8 MAR units, but beyond that, the performances of all three configurations are equally low (Figure 14 right). However, in Kalu Ganga, the performance of Configuration 2 is distinctly better than that of Configuration 3 for all storage capacities, unlike in Malwatu Oya where there is little discrepancy between the two configurations. The results illustrate that the optimum reservoir arrangement is unique to each river; i.e., while distributed small reservoirs work best in some river basins, a combination of distributed larger reservoirs may prove to be optimum in others. When the cumulative storage capacity is at the MAR, Configuration 3 is the lowest performer similar to the Malwatu Oya basin.

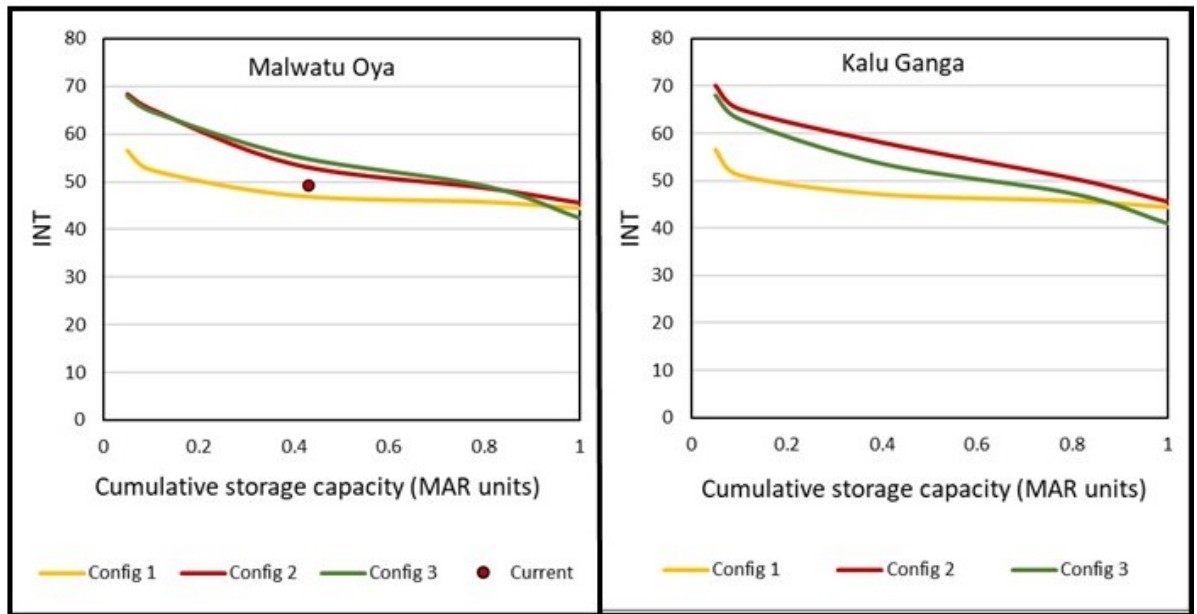

**Figure 14.** Variation of the integrated index (INT) value with cumulative storage capacity across reservoir system Configurations 1, 2 and 3 in Malwatu Oya and Kalu Ganga.

Table 6 shows alternative ways in which the integrated index was computed (INT1 to INT4) to assess its sensitivity to the number and type of indicators combined and the weights assigned. INT1 to INT3 each combine only one of the ecology-related indicators (EF Yield, MRRI or $MRCI_{internal}$) with equal weights assigned to economic benefits, ecological effects and social effects. In contrast, INT4 includes an average of all three ecology-related indicators with equal weights assigned to the three components economic benefits, ecological effects and social effects. Similar to INT, these alternative indices were also converted to a percentage score based on their scores out of a maximum of 45 (considering 15 different storage scenarios for both basins excluding the current storage scenario of Malwatu Oya). The behavior of the alternative integrated indices in comparison with INT is shown in Figure 15.

**Table 6.** Combinations and weights of individual indices to form alternative integrated indices.

| Alternative Integrated Index | Estimation Formula |
|---|---|
| INT1 | $\frac{WS\ Yield + EF\ Yield + NE}{45} \times 100$ |
| INT2 | $\frac{WS\ Yield + MRRI + NE}{45} \times 100$ |
| INT3 | $\frac{WS\ Yield + MRCI_{internal} + NE}{45} \times 100$ |
| INT4 | $\frac{WS\ Yield + \frac{1}{3}(EF\ Yield + MRRI + MRCI_{internal}) + NE}{45} \times 100$ |

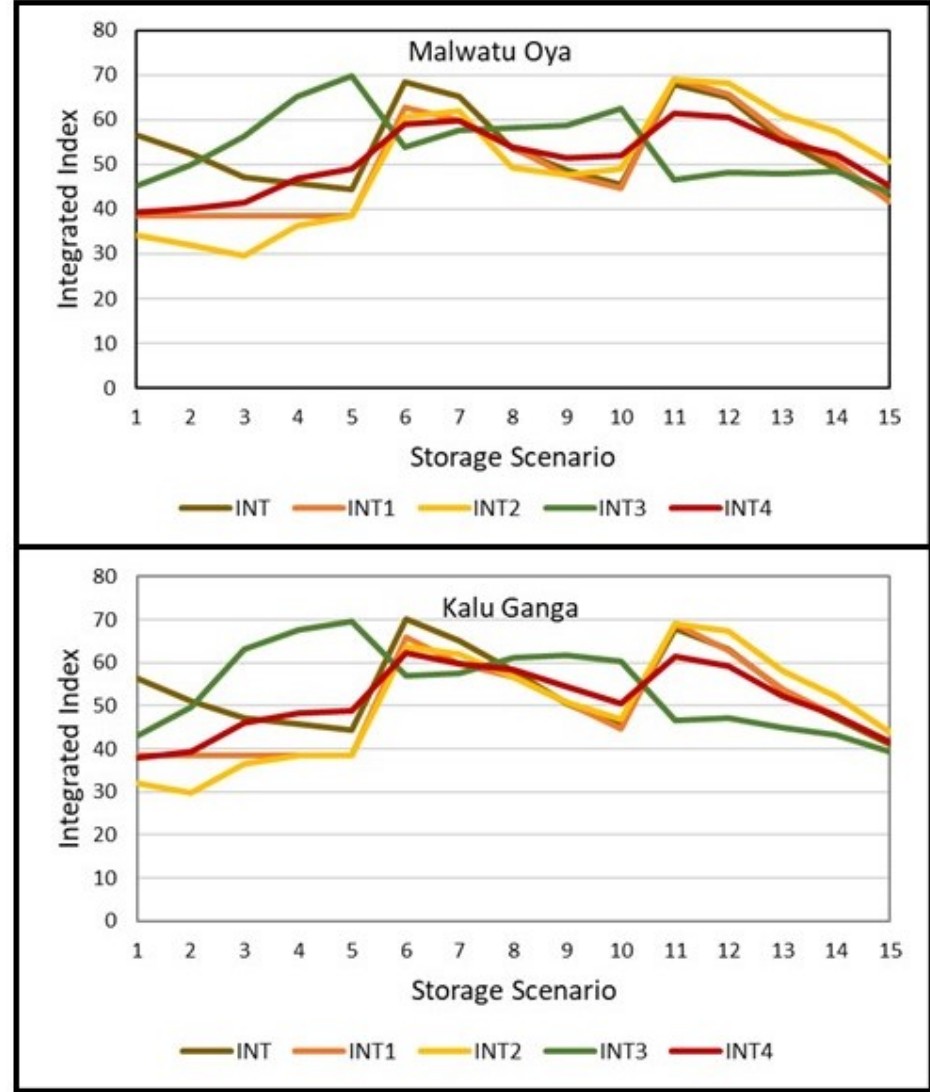

**Figure 15.** Behavior of alternative integrated indices in comparison with INT for the Malwatu Oya and Kalu Ganga basins.

All alternative indices except INT3 exhibit similar behavior along the storage scenarios (Figure 15). The behavior of INT3 contrasts those of the other alternative integrated indices. INT3 is formed by combining WS Yield, $MRCI_{internal}$ and NE. In INT3, ecosystem aspects are only represented by the degree of internal connectivity ($MRCI_{internal}$), while implications of flow regulation or the quantity of available EF are not considered. Therefore, storage scenarios 1 to 5, which have the highest internal connectivity, perform better than other scenarios.

The reservoir configurations, indicators and weights for this research were selected for illustrative purposes and represent only a subset of all possible alternatives. The approach illustrated in this paper is generic and can be used in planning alternative storage scenarios in river basins, either in expanding existing storage capacity (e.g., in a basin like Malwatu Oya) or in developing new storage in unregulated basins (e.g., in a basin like Kalu Ganga). However, it is important to appropriately identify, prioritize and weigh individual indicators depending on the specific requirements of each river basin in consultation with a wide group of stakeholders, a prerequisite illustrated by the results for different combinations of indicators in Figure 15.

## 5. Conclusions

This paper is the second of two connected papers illustrating tools and approaches for basin-wide planning of surface water storage development and presents an approach to comparing alternative reservoir arrangements in a river basin based on the measurement of economic benefits and ecological and social effects. Proxy indicators representing economic benefits and ecological and social effects were identified, and their behavior in varying reservoir system configurations was investigated in two example river basins in Sri Lanka. Alternative indicators and methods (built on existing ones) were derived to enhance differences between ecological and social impacts of different reservoir arrangements and to overcome limitations in data.

Modified internal and external river connectivity indices derived in the research illustrate that if a single new reservoir is introduced to a previously fully connected network, the lowest internal and external connectivity is generated if this new reservoir is placed on the main stem between the two locations at which the normalized upstream basin area is approximately equal to 0.5 and 0.75. Results also show that the impact on internal connectivity is higher as the first few reservoirs are added to the network but this impact diminishes as the number of reservoirs increases. Approximations to estimate normalized externality (NE), a measure of equity of access to stored water, show that the ratio of the total number affected to the total number of beneficiaries increases as storage capacity expands in river basins. Results also show that at a given storage capacity, distribution of access to stored water differs between different types of reservoir system configurations; i.e., the more spatially distributed the location of storage infrastructure, the more equitable water access will be whether there are distributed small reservoirs or concentrated large reservoirs.

Integrated results of indicator behavior suggest that, in general, distributed small reservoirs perform better than concentrated large ones. However, this difference in performance is not apparent at large cumulative storage capacities (e.g., when storing above 80% of the MAR of the basin). Therefore, accumulated effects (both positive and negative) of a large number of small reservoirs may still be as significant as those of a small number of large reservoirs at very large aggregated storage capacities in river basins. The results further suggest that the optimum reservoir arrangement is unique to each river; i.e., while distributed small reservoirs work best in some river basins, a combination of larger reservoirs may prove to be optimum in others.

The approach presented in this paper is aimed at formulating a preliminary plan for the integrative assessment of artificial surface storage options in a river basin to maximize benefits and minimize ecosystem and social costs. While this research illustrates a limited number of reservoir configurations and indicators, a future research direction that may be pursued is the inclusion of a larger range of reservoir configurations and indicators and the use of analytical techniques such as multiobjective optimization to identify optimal configurations. Moreover, this research presents the first applications of modified indicators on river regulation, connectivity and equity. These indicators can be applied in different river basins to analyze their behavior in more detail (including uncertainty and sensitivity) so they can be adapted to individual circumstances. However, it is important that a wide group of stakeholders are consulted in identifying, prioritizing and weighing appropriate

indicators and potential reservoir arrangements depending on the specific requirements of each river basin.

**Author Contributions:** Conceptualization, N.E., V.S. and L.U.; Methodology, N.E.; Formal Analysis, N.E.; Investigation, N.E.; Writing—Original Draft, N.E.; Writing—Review and Editing, V.S. and L.U.; Supervision, V.S. and L.U. All authors have read and agreed to the published version of the manuscript.

**Funding:** This research was supported by the International Water Management Institute (IWMI) as part of the CGIAR Research Programs on Climate Change Agriculture and Food Security (CCAFS) and Water Land and Ecosystems (WLE).

**Institutional Review Board Statement:** Not applicable.

**Informed Consent Statement:** Not applicable.

**Data Availability Statement:** Simulated runoff data used in the analysis can be made available on request from the authors.

**Acknowledgments:** The authors are grateful to Matthew McCartney, Research Group Leader, Sustainable Water Infrastructure and Ecosystems (IWMI), for his constructive feedback on the manuscript. The authors would also like to thank Lal Mutuwatte (IWMI) and Madusanka Thilakarathne (Consultant, IWMI) for their role in developing the SWAT models.

**Conflicts of Interest:** The authors declare no conflict of interest.

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
