# Peer review of "Sustainable Surface Water Storage Development: Measuring Economic Benefits and Ecological and Social Impacts of Reservoir System Configurations"

_water, doi:10.3390/w14030307_

Round 1
Reviewer 1 Report
Sustainable Surface Water Storage Development: Measuring Economic Benefits and Ecological and Social Impacts of Reservoir System Configurations
By Nishadi Eriyagama, Vladimir Smakhtin and Lakshika Udamulla
This is an interesting paper that tries to assess the economic, ecological and social impacts of different reservoir configurations in a river basin. What makes this paper for me special, is that it further develops an idea that I and my colleague Joyeeta Gupta proposed more than 10 years ago (Van der Zaag and Gupta, 2008).
I may thus be biased, but I nevertheless am of the opinion that this paper makes an interesting contribution to our thinking about the impact of dam configurations in river basins. I however do have a number of comments.
- The paper assumes a water yield with a 100% reliability (line 83), which I found a bit odd. Normally a percentage in the range of 95-99% is used, but never 100%, as you a 100% assurance level is normally uneconomic. But of course the “real” assurance is determined by the length of the time series, and in this case this is limited. Implying that in this case an assurance of supply of 100% does not mean that it will always be achieved.
- The Environmental Flow EF as defined in this paper (in section 2.2.2) is a very crude indicator, as it is measured over one entire hydrological year. It may thus hide features of a modified flow regime that are key for ecological processes, such as minimum flows during the low flow season. The crudeness of this indicator should be explicitly stated. Having said this, I must add that I found it very good that an environmental indicator was included in this paper.
- It is stated that the “distribution factor” is an indicator with a value between 0 and 1 (line151), but the way this is written may be misconstrued to mean that the MRRI value must fall within this range. But this is not so; the MRRI, which can have a value larger than 1, if I am not mistaken. (Similarly, the “cumulative storage capacity in basin”, as mentioned in Table 2, can have a value larger than 1.0 MAR units, but it is suggested in this table that it can’t. This I don’t understand.)
- The DCI and MRCI indicators (section 2.2.3). I failed to understand what the meaning is of the external DCI (or MRCI) and the internal DCI (or MRCI), and what their difference is. Why should the internal DCI be based on the square of the length of river segments (and likewise the MRCI on the squares of the basin areas of those segments)? What is the physical meaning?
- The NE indicator. I found equation 9 problematic. Why aren’t the number of benefactors of a dam not simply equated to the number of people living upstream of that dam (piup)? Why should it be weighted by the relative size of the dam (piup is multiplied by the DOR or residence time, namely (si/qi))? Note that in equation 9, the dimension of nb has a time unit (see my comment below)! I am not sure what the implications are of equation 9 for the final results, but this should be better justified.
I like the attempt to apply the indicators to the two cases in Sri Lanka.
- Section 4.1, lines 374-376, read “the cumulative WS Yield from a network of reservoirs declines with increasing levels of spatial distribution of reservoirs even though the aggregated storage capacity remains the same.” Yes, but also note that with increasing levels of spatial distribution, the spatial unequal distribution of access to the benefits of dam water decreases; so more people will be able to use and benefit from the water yield.
- Section 4.4; Figure 12: this figure is rather suggestive if you would simply interpret this figure in terms of the X-axis: the more storage, the more the two configurations converge. The point, however, is different: at a given total storage, the NE (i.e. the distribution of water access) for the three configurations is very different; and then you can see that for Malwatu Oya, for a cumulative storage of 0.4 MAR, Config 3 has a much more equal distribution (or a much lower NE) than for Config 2 (or for Config 1, which is not shown in the graph, but goes beyond the Y-scale).
- The integrated index INT is introduced in section 4.5, but could have been more adequately introduced, and defined, in the methodology section. Now it comes out of the blue, including that alternative INT indices are defined and calculated. To me the criteria used, and their logic, to vary the INT indices was not explained and not clear to me. It would be preferable to identify only two or three different types of INT, well defined and justified, and then present the results and analyse these.
- Section 5, conclusions. It is concluded in the 2nd paragraph that “the ratio of the total number affected to the total number of beneficiaries increases as storage capacity expands in river basins.” This is probably a correct conclusion. But what is also important to conclude is that the more spatially distributed the location of storage infrastructure is, the more equitable water access will be.
Three detailed comments:
- It is for me very important that correct units/dimensions are used for the indicators. There is a fundamental difference between stocks (stores) and flows (or fluxes). The former have as a unit: volume (or L3) and the latter volume per time unit (L3/t]. This may look futile but it isn’t. One consequence is that you cannot just add up stocks and fluxes. A second is that if you divide a stock by a flux, you get an indicator with an interesting unit: time! And that is exactly what the DOR [Degree of Flow Regulation] is, which is better known as (and more aptly and meaningfully captured by) residence time.
As a consequences, units written in lines 313, 314,326, 327, 369 (Table 1), 562 (Table 5) need to be corrected.
I prefer that for all indicators used their units are explicitly declared (they all seem unit-less, except for DOR, RRI and MRRI, if I am not mistaken).
- I detected eight references mentioned in the text that do not appear in the reference list, namely Hanaksaki et al., 2018; Jager et al., 2015; Roozbahani et al., 2017; Thomas and Burden, 1963; Rodriguez-Iturbe et al., 2009; Panabokke et al., 2002; Nandalaland Ratnayake, 2010; Atukorala, 2006.
- Some typos: Line 62, first word needs to be capitalised. Line 272: we’re -> were
Reviewer 2 Report
This paper tries to find the most suitable alternative reservoir arrangements in a river basin based on the economic benefits and socio-ecological impacts.
This special issue is about Water Energy Nexus. So the relevancy for this paper with respect to the special issue need to be properly elaborated. The structure of the paper need to be modified. It must follow Introduction > Scope and Justification > Novelty > Literature Surveys > Methods Used > Detail Methodology > Results > Discussions > Conclusions.
The justification of selection of the two locations need to be explained.
Whether the selected indicators were arbitrarily picked or has some logic. If there is selection procedure then elaborate it properly.
Whether any Uncertainty Analysis and Sensitivity analysis conducted ? If not required why ?
Although the study is innovative and nicely written but to become a reliable scientific document it need to improve by incorporating the above points. Each and every conclusion must be supported with explanations and a reference to similar studies.
